# Assembly of the mitochondrial outer membrane module of the trypanosomal tripartite attachment complex

**Philip Stettler[1,2], Salome Aeschlimann[1], Bernd Schimanski[1], André Schneider●[1]\***

**1** Department of Chemistry, Biochemistry and Pharmaceutical Sciences, University of Bern, Bern, Switzerland, **2** Graduate School for Cellular and Biomedical Sciences, University of Bern, Bern, Switzerland

\* andre.schneider@unibe.ch

## Abstract

The parasitic protozoan *Trypanosoma brucei* has a single mitochondrial nucleoid, anchored to the basal body of the flagellum via the tripartite attachment complex (TAC). The detergent-insoluble TAC is essential for mitochondrial genome segregation during cytokinesis. The TAC assembles *de novo* in a directed way from the probasal body towards the kDNA. However, the OM TAC module which is composed of five subunits, has previously been suspected to follow more complicated assembly pathways. Here, we identified four detergent-soluble OM TAC module subcomplexes that we assign to two classes. One class contains an oligomeric TAC40 complex that according to AlphaFold contains 6–8 subunits, as well as two subcomplexes of different sizes comprising TAC40, TAC42, and TAC60. The second class consists of a single complex composed of TAC65 and pATOM36. We show that the two subcomplex classes form independently and accumulate upon impairment of TAC assembly. The expression of an N-terminally truncated TAC60 variant causes the accumulation of the larger TAC40/TAC42/TAC60 complex and blocks completion of OM TAC module assembly. This suggests the following assembly pathway: i) TAC40 oligomerizes, ii) TAC42 and TAC60 bind the TAC40 oligomer forming two discrete larger intermediates, where iii) the larger subcomplex merges with the pATOM36/TAC65 subcomplex subsequently forming the OM TAC module.

## Author summary

Mitochondria are essential for nearly all eukaryotes. This requires proper segregation of the organelles and their genomes to the daughter cells during cell division. For trypanosomes and their relatives this represents a unique challenge because they have only a single highly compacted mitochondrial nucleoid, containing the bipartite mitochondrial genome, known as the kDNA. To ensure

**Data availability statement:** All relevant data are within the paper and its Supporting Information files.

**Funding:** This study was supported in part by project grant SNF 205200 to A.S. and by a grant of the NCCR RNA & Disease, a National Centre of Competence in Research (grant number 205601) to A.S both funded by the Swiss National Science Foundation (https://www.snf.ch/en). The funders had no role in study design, data collection and analysis, decision to publish, or preparation of the manuscript.

**Competing interests:** The authors have declared that no competing interests exist.

faithful segregation, kDNA replication is tightly coordinated with the nuclear cell cycle and the kDNA is physically connected to the flagellar basal body via the tripartite attachment complex (TAC), which spans both mitochondrial membranes. The TAC consists of at least nine subunits and its biogenesis arguably implies the most extreme lateral sorting event known for any mitochondrion. Five subunits, four integral and one peripheral outer membrane protein, make up the outer TAC module. Using blue native PAGE and RNAi analyses we have identified four soluble subcomplexes that likely represent assembly intermediates of the outer membrane TAC module. These results suggest that the outer membrane TAC module is assembled following a branched pathway that includes an unusual ring-like intermediate of 6–8 subunits of a trypanosomatid-specific VDAC-like beta-barrel protein.

## Introduction

Genome replication and faithful segregation of the replicated genomes to daughter cells during cell division are arguably the most central processes of life. The most complex situation is found in eukaryotes which have a nuclear genome and additional organellar genomes.

How replicated organellar genomes are segregated during organellar fission processes is an understudied subject. Mitochondrial genome segregation processes have mainly been studied in *Saccharomyces cerevisiae* and humans [1–3] which both belong to the eukaryotic supergroup of the Opisthokonts [4]. Both species contain many mitochondria (in human up to several hundred per cell) that form highly dynamic networks which are constantly remodeled by fission and fusion processes [5–7]. The mitochondrial genome is organized in nucleoids, each consisting of a few copies of the mitochondrial genome associated with numerous DNA binding and other proteins [8,9]. Nucleoids outnumber mitochondria, appear to be associated with the mitochondrial inner membrane (IM) and can be transported actively within the network. The processes ensuring proper segregation of nucleoids during mitochondrial fission appear to involve mitochondria-associated ER domains but how they work in detail remains to be elucidated [10–12].

In the present study we were studying mitochondrial genome segregation in the parasitic protozoan *Trypanosoma brucei* which belongs to the Discoba group [4]. *T. brucei* is an experimentally highly accessible model system and has arguably the best studied mitochondrion outside the Opisthokonts [13–21].

*T. brucei* and its relatives, most of which are parasites, are famous for having a single mitochondrion with a single nucleoid only [22]. The structure of the trypanosomal mitochondrial genome, termed kinetoplast DNA (kDNA), is very complicated. It consists of two genetic elements: approximately 25 maxicircles (22 kb in length) and approximately 5'000 minicircles (1 kb in length). Maxicircles and minicircles are highly concatenated among themselves and between each other forming a large disk-shaped kDNA network [19,20,23]. Maxicircles encode mainly subunits of the

respiratory complexes [24]. Many of their genes represent cryptogenes, thus their transcripts must be edited by multiple uridine insertions and/or deletions to become functional mRNAs. Minicircles, on the other hand, encode guide RNAs that provide the information for RNA editing [18,25,26].

The kDNA is permanently tethered, across the two mitochondrial membranes, with the basal body of the flagellum by a unique structure called the tripartite attachment complex (TAC) (S1 Fig) [27–29]. The function of the mega-Dalton-sized TAC is to allow the coupled segregation of the single kDNA disk and the single flagellum [27,30]. This implies that the kDNA network needs to be replicated in coordination with the nuclear cell cycle [19,22].

In *T. brucei* nine TAC subunits each present in several hundred to a few thousand copies have been identified so far [30,31] (S1 Fig). The TAC subunits and the TAC architecture are conserved within Kinetoplastids [19,31]. Traditionally, the TAC has been divided into three morphological domains based on transmission electron microscopy: the cytosolic exclusion zone filaments, the differentiated mitochondrial membranes, and the unilateral filaments in the matrix [28]. However, characterization of the nine TAC subunits, which likely represent the nearly complete set of TAC components, now allows to define three TAC modules based on their molecular composition: the cytosolic, the outer membrane (OM), and the inner TAC modules [31] (S1 Fig).

The "cytosolic TAC module" connects the basal body to the OM TAC module and consists of the single subunit p197 [32,33]. The C- and N-termini of p197 interact with the basal body and TAC65 of the OM module, respectively. The large central part of p197, making up approximately 84% of the protein, consists of 35 or more near-identical α-helical repeats of 175 aa in length [34] and determines the distance between the basal body and the OM [32]. Thus, the predicted molecular mass of p197 is more than 880 kDa making it the largest protein of *T. brucei*.

The "OM TAC module" consists of the five subunits TAC65, pATOM36, TAC40, TAC42 and TAC60 [31]. The globular TAC65 is a peripheral OM protein which faces the cytosol [35,36]. It interacts with the N-terminus of p197 [32] and binds to pATOM36, one of four integral OM proteins of the OM module [36]. pATOM36 has a dual function, it is an essential subunit of the TAC structure, whereas outside of the TAC it mediates biogenesis of the atypical protein translocase of the OM (ATOM) complex [36]. Reciprocal complementation experiments between yeast and *T. brucei* have shown that pATOM36 and the yeast MIM complex have identical functions in the biogenesis of a subset of OM proteins [37]. TAC40 and TAC42 are β-barrel membrane proteins that form a complex with TAC60 which has two α-helical transmembrane domains [38,39]. Both the N- and C-termini of TAC60 face the cytosol, but in contrast to the N-terminus, the cytosolically exposed C-terminus of TAC60 is dispensable for TAC function [38]. The 142 aa long intermembrane (IMS) -exposed loop of TAC60 contains the binding site for the interaction with the C-terminus of p166 [40].

p166 is a subunit of the "inner TAC module" and contains a single transmembrane domain close to its C-terminus [31,41,42]. The IMS-exposed loop of TAC60 binds to the IMS-exposed C-terminus of p166 and forms a stable contact site between the OM and IM. It was shown that the minimal p166 binding site of TAC60 consists of a short kinked α-helix that via hydrophobic interactions binds to the C-terminal α-helix of p166 [40]. The large soluble domain of p166 is exposed to the matrix and binds to TAC102, a soluble matrix protein [41,43,44]. While TAC102 localizes close to the kDNA it does not bind to it directly. Four proteins, TAC53, TAP110, KAP68, and mtHMG44, which localize between TAC102 and the kinetoplast have been identified [45–47]. Of these only TAC53 behaves like a classical TAC subunit [47]. KAP68 can bind to DNA at least in vitro [45]. However, what precise functions these proteins have and how they are arranged relative to each other is not known. Thus, how exactly the TAC is anchored to the kDNA remains unclear [31].

An interphase trypanosomal cell has a single flagellum and a single kDNA that is connected to the TAC [28]. The coordinated assembly and duplication of the TAC during the cell cycle represents one of the most extreme sorting and assembly events seen in any mitochondria [30,31]. It is known that the overarching principle of TAC formation is based on a *de novo* and hierarchical assembly of its subunits [35]. Starting with p197 at the pro basal body the assembly proceeds towards the kDNA. Generally, depletion of basal body-proximal TAC subunits results in the delocalization of all distal subunits. However, there are hints that the complex OM TAC module behaves differently. Although TAC65 is the

direct interaction partner of p197 and a cytosolically exposed peripheral OM protein, it cannot assemble into the TAC after ablation of TAC40 or TAC60 [35].

Using a combination of RNAi cell lines and blue native (BN)-PAGE analyses we have studied the biogenesis of the OM TAC module. We could show that it involves the independent formation of various detergent-soluble assembly intermediates consisting of either TAC40/TAC42/TAC60 and TAC65/pATOM36, respectively.

## Results

### OM TAC module subunits form detergent-soluble subcomplexes

The fully assembled TAC structure is insoluble in non-ionic detergents [32,41,42]. However, proteomic analysis of pull-down experiments of digitonin-solubilized crude mitochondrial fractions using the tagged OM TAC subunits TAC40, TAC42, TAC60, TAC65 and pATOM36 suggested the existence of two main classes of detergent soluble subcomplexes consisting of either (i) TAC40, TAC42 and TAC60 or (ii) TAC65 and pATOM36, respectively [36,38].

To analyze the pulldown experiments in more detail, we pooled all five data sets [36,38] to generate an edge-weighted protein-protein interaction network visualizing the digitonin-soluble interactomes of TAC40, TAC42, TAC60, TAC65 and pATOM36 (Fig 1A). This network confirmed the two previously suggested soluble OM TAC module subcomplex classes mentioned above (Fig 1A, blue and red). Notably, there are no connections between the two subcomplex classes, except for a non-reciprocal low enrichment of TAC60 in the pATOM36 pulldown. The only notable additional proteins recovered in more than one pulldown experiment are POMP19 and J31, subunits of a quality control pathway connected to α-helically anchored OM proteins [48], and an uncharacterized metallopeptidase (Tb927.9.13490), a known contaminant detected in previous unrelated pulldown experiments [49–51]. We would like to emphasize that pull down experiments allow to define the composition of protein complexes but do not provide information which subunits directly interact with each other.

Next, we decided to investigate the detergent-soluble protein complexes of the OM TAC module in the procyclic form (PCF) of T. brucei in more detail using BN-PAGE and immunoblot analyses, with the ultimate aim to gain insight into its assembly process (Fig 1).

Using a polyclonal antiserum recognizing TAC40, we detected three different TAC40-containing subcomplexes with estimated sizes of ~535, ~770, and ~920 kDa, respectively (Figs 1B, left panel, S2 Fig). The bands corresponding to the ~535 and ~770 kDa subcomplexes contained similar amounts of TAC40, whereas the band corresponding to the ~920 kDa subcomplex contained much less of the protein, suggesting the subcomplex is less abundant. Alternatively, it is also possible that the blotting of the larger complexes is less efficient.

To detect the OM TAC module subunits TAC42 and TAC60, we prepared transgenic cell lines expressing C-terminally HA or myc-tagged versions of the proteins (Fig 1B, middle and right panels). Immunoblot analyses showed that tagged TAC42 and TAC60 were exclusively detected in the ~770 and ~920 kDa TAC40-containing subcomplexes, respectively.

Thus, our results confirm the previous immunoprecipitations experiments [38] and show that TAC40 is mainly present in three distinct subcomplexes. The lowest one of ~535 kDa consists exclusively of TAC40 whereas the ~770 and ~920 kDa subcomplexes likely in addition contain variable amounts of TAC42 and TAC60. Moreover, the edge-weighted protein-protein interaction network shown in Fig 1A indicates that no other proteins are stoichiometrically associated with these subcomplexes.

Reciprocal immunoprecipitations have shown that TAC65 interacts with pATOM36 and vice versa [36]. BN-PAGE analysis of cell lines expressing myc-tagged TAC65 revealed a diffuse TAC65-containing complex of ~440 kDa (Fig 1B, left panel). An analogous experiment indicated that HA-tagged pATOM36 appears to be present in the same ~440 kDa subcomplex but also in a smaller subcomplex migrating at approximately ~100 kDa (Fig 1B, right panel). The latter subcomplex was reported previously [36,52]. The fact that this subcomplex is also detected when pATOM36 is expressed in S. cerevisiae shows that it represents the fraction of pATOM36 that functions in OM protein biogenesis [37]. Thus, the ~440 kDa complex, likely consisting of TAC65 and pATOM36, is the one relevant for assembly of the OM TAC module. The signals near the top of the gel represent complexes that accumulate at the stacking gel-separating gel interface.

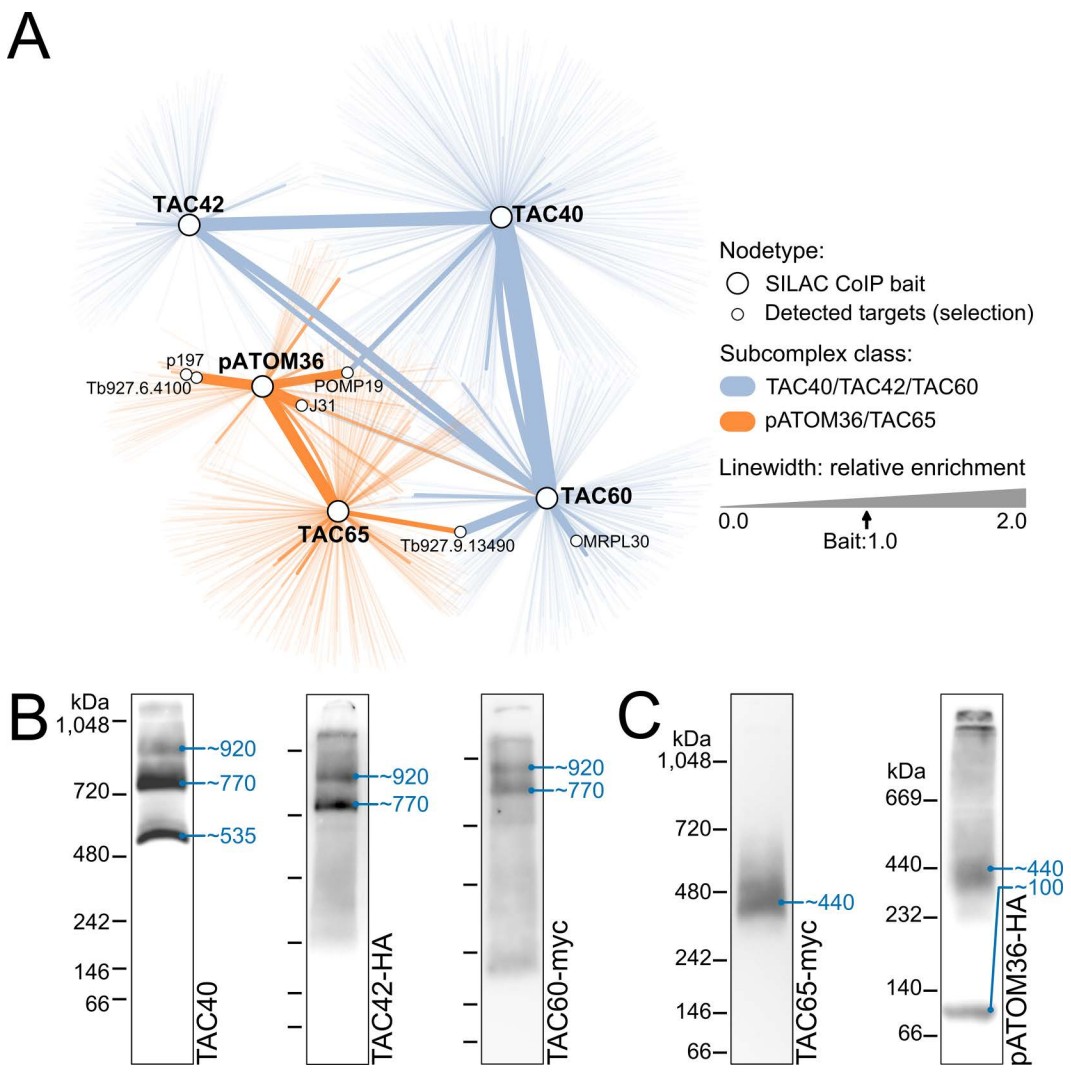

**Fig 1. OM TAC module subunits form detergent-soluble subcomplexes. (A)** Edge-weighted protein–protein interaction network generated using Cytoscape (version 3.9). The network depicts all proteins enriched (mean foldchange > 1.0) in previously published pulldown experiments of TAC40, TAC42, TAC60, TAC65, and pATOM36. Edge widths correspond to the relative enrichment (mean fold change target/mean fold change bait). Edge colors mark interactions of the TAC40/TAC42/TAC60 subcomplex class (blue) and the pATOM36/TAC65 subcomplex class (orange). Notable targets are annotated. **(B)** Immunoblots of BN-PAGE experiments probed for TAC40, TAC42-HA, or TAC60-myc with protein (TAC40) or tag specific (TAC42, TAC60) antibodies. Figures display uncropped blots including the pseudo stacking gels. The positions of marker proteins with their size in kDa are indicated on the left of each lane. Based on these markers, the sizes of the detected protein complexes were estimated (blue numbers) (see Material and methods). **(C)** As in (B) but immunoblots were probed for TAC65-myc and pATOM36-HA with tag specific antibodies.

In summary, our results are in line with previous analyses [36,38] and show that TAC65 together with pATOM36 is present in a single protein subcomplex of ~440 kDa, whereas pATOM36 in addition forms an ~100 kDa subcomplex.

## Depletion of OM TAC module subunits alters subcomplex formation

To investigate the effects of depletion of OM TAC module subunits on subcomplex compositions, we used a combination of RNAi, BN-PAGE, and immunoblot analyses. The left panel of Fig 2A shows that depletion of either TAC60 or TAC42 prevents the formation of the larger TAC40-containing subcomplexes (~770 and ~920 kDa) and concomitantly causes the

accumulation of the smaller ~535 kDa subcomplex containing only TAC40. This suggests that neither a TAC40/TAC60 nor a TAC40/TAC42 pair can independently form discrete complexes detectable by BN-PAGE in the absence of the third subunit. Similarly, depletion of either TAC40 or TAC60 eliminates the two discrete TAC42-containing subcomplexes (~770 and ~920 kDa) (Fig 2A, middle panel). Instead, TAC42 appears to be dispersed into a poorly resolved smear representing complexes with a large spread of molecular weights. This indicates that TAC42 alone cannot form defined subcomplexes with either TAC40 or TAC60. Additionally, unlike TAC40, TAC42 does not assemble into a stable oligomeric subcomplex. TAC60 behaves differently: in the absence of TAC42, it forms very large subcomplexes of >1'000 kDa, while after TAC40

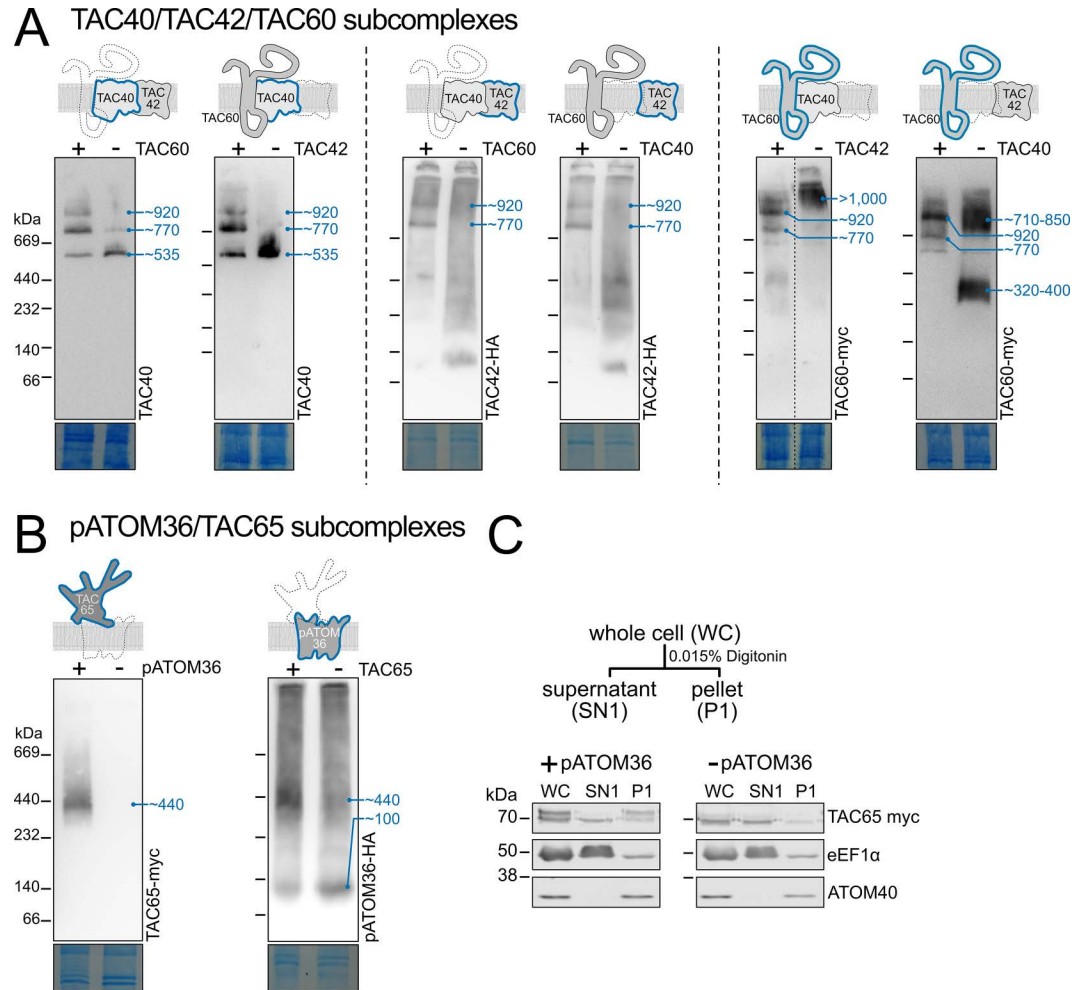

**Fig 2. Depletion of OM TAC module subunits alters subcomplex formation.** (A) Left panel: Immunoblots of BN-PAGE gels were probed with a TAC40 antiserum. Inducible RNAi knock down cell lines for TAC60 or TAC42 were analyzed. The approximate sizes in kDa of detected subcomplexes are indicated with blue numbers. Schematics on top indicate which TAC subunits were detected by the immunoblot (blue) and which TAC subunits were the targets of the knockdown (dashed line). Presence ('+') and absence ('-') of the TAC subunits in question are indicated at the top of each gel. Bottom panels show sections of Coomassie blue-stained gels and serve as loading controls. Immunoblots of BN-PAGE gels shown in all following panels and figures are presented in an analogous way. (B) as in (A) detected proteins and targets of depletion are indicated. (C) Top panel: Scheme depicting the one-step 0.015% digitonin fractionation. Bottom panel: Immunoblots of an SDS-PAGE gel of a one-step digitonin fractionation of cells expressing TAC65-myc in the presence of (left) or depleted for (right) pATOM36. Eukaryotic elongation factor 1 alpha (eEF1α) and the atypical translocase of the outer membrane 40 (ATOM40) serve as cytosolic and mitochondrial markers, respectively.

depletion, it is found in two discrete groups of subcomplexes of ~320–400 kDa and of ~710–850 kDa, respectively (Fig 2A, right panel). The composition of these aberrant TAC60-containing subcomplexes remains unknown.

In summary, our analysis reveals that TAC40, TAC42, and TAC60 form three subcomplexes: one of ~535 kDa consisting of a TAC40 oligomer and two of ~770 and ~920 kDa, respectively, which are likely exclusively composed of various amounts of TAC40, TAC42 and TAC60.

Fig 2B shows that pATOM36 and TAC65 are both present in a ~440 kDa complex. Depleting either protein causes the dissociation of this complex, suggesting that it consists solely of pATOM36 and TAC65 (Fig 2B). As expected, the ~100 kDa pATOM36 complex, which functions in OM protein biogenesis, still assembles in the absence of TAC65. However, TAC65 alone does not form a complex, and no monomeric TAC65 signal is detected in crude mitochondrial extracts analyzed by BN-PAGE. The bottom right panel of Fig 2C further reveals that after pATOM36 depletion, TAC65 levels are drastically reduced in the crude mitochondrial pellet fraction obtained by 0.015% digitonin extraction. This is expected, as TAC65 is a peripheral OM protein without predicted transmembrane domains, and its association with the OM depends on pATOM36. Note: similar to TAC60, TAC65 appears as a double band on denaturing gels [36]. The upper band likely results from an unknown post-translational modification and serves as a marker for proper TAC integration [38] (Fig 2C, bottom left panel).

## The two classes of OM TAC module subcomplexes form independently

As shown above the OM TAC module subunits form two classes of detergent-soluble subcomplexes. The first class consists of a TAC40 oligomer of ~535 kDa and two subcomplexes (~770 and ~920 kDa) that contain various amounts of TAC40, TAC42 and TAC60. The second class includes a ~440 kDa complex composed of pATOM36 and TAC65 as well as the OM protein biogenesis complex of ~100 kDa that contains pATOM36.

According to the hierarchical model of TAC formation, the different TAC subunits are expected to assemble in a strict stepwise manner, beginning at the (pro)basal body and extending toward the kDNA [35]. Based on this model, the more basal body-proximal 440 kDa subcomplex, consisting of pATOM36 and TAC65, should be required for the formation of the three more basal body-distal TAC40-, TAC42- and TAC60-containing subcomplexes (~535 kDa, ~770, ~920 kDa).

However, our results contradict this expectation. Even after depletion of pATOM36 or TAC65, resulting in the absence of the ~440 kDa subcomplex, the TAC40-, TAC42-, and TAC60-containing subcomplexes (~535 kDa, ~770 kDa, and ~920 kDa) still form (Fig 3A). In addition, depletion of the TAC40-containing subcomplexes does not disrupt the more basal-body-proximal ~440 kDa subcomplex composed of pATOM36 and TAC65 (Fig 3B).

These findings are surprising, as previous immunofluorescence analysis has shown that depletion of any of the tested OM TAC module subunits (TAC40, TAC60, TAC65) leads to the dispersion of the entire module [35]. Despite this, the subunits are not degraded, as they remain detectable on denaturing gels [35]. Instead, their dispersal across the OM prevents their detection via immunofluorescence.

## TAC OM module subcomplexes form independently of cytosolic and inner TAC modules

The cytosolic and the inner TAC modules are both anchored at the OM TAC module. Thus, we examined the fate of OM TAC subcomplexes upon depletion of either the cytosolic TAC module alone or both the cytosolic and inner TAC modules simultaneously. In contradiction to the hierarchical assembly model, depletion of the cytosolic TAC module subunit p197, which connects to the OM TAC module subunit TAC65, did not disrupt the formation of either class of OM TAC module subcomplexes (Fig 4).

Instead, we saw a time-dependent accumulation of the two TAC40-, TAC42-, and TAC60-containing subcomplexes (~770 and ~920 kDa). Intriguingly, it appeared that the ~770 kDa subcomplex accumulated faster than the ~920 kDa subcomplex and that its levels declined at later time points of induction. This was not just the case for the mean values of the triplicate experiments (Fig 4A, left graph) but the same was also seen in each of the three individual experiments included

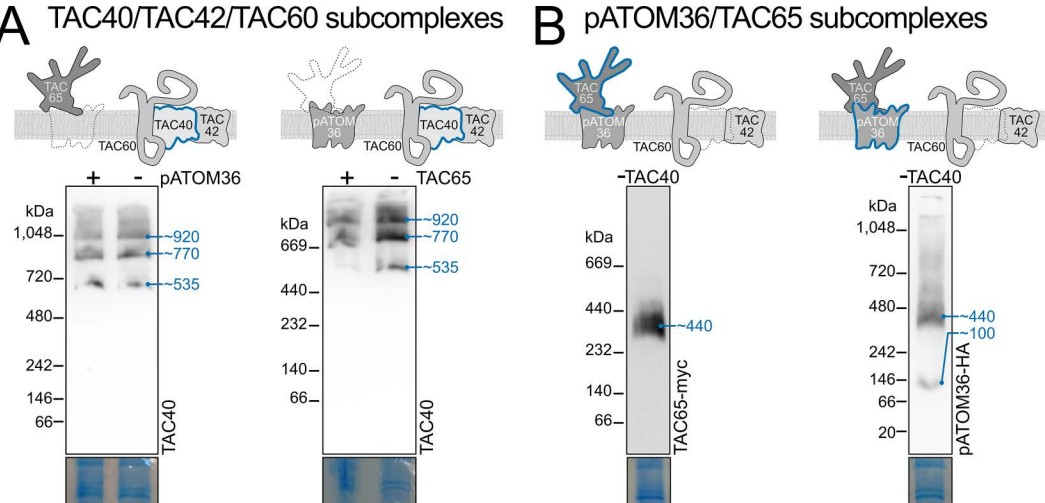

**Fig 3. The two classes of OM TAC module subcomplexes form independently. (A)** Immunoblots of BN-PAGE experiments probed for TAC40 of cell lines with inducible pATOM36 (left) or TAC65 (right) knockdown. **(B)** As in (A) but immunoblots of TAC40 depleted cells probed for TAC65-myc (left) and pATOM36-HA (right) with anti-tag antibodies are shown.

in the analysis. However, the levels of the smallest subcomplex of ~535 kDa consisting of a TAC40 oligomer remained constant (Fig 4A, left panel). The same time course experiment was also performed in cell lines simultaneously depleted for both the cytosolic and inner TAC modules and similar kinetics were observed (Fig 4A, right panel). Although in this case it was less clear whether the ~770 kDa subcomplex accumulated faster than the ~920 kDa subcomplex.

However, there are some caveats to be considered in the interpretations of the experiments shown in Fig 4A. Comparisons of the data points from triplicate experiments before and after RNAi induction using ANOVA followed by *post hoc* Student's t-tests and the Holm-Bonferroni method resulted in adjusted *P* values <0.1 but did not always reach the statistical significance of <0.05. Nevertheless, because the same quantitative trends were observed in both sets of experiments (n = 3 each) done in two different cell lines (Fig 4, left and right panels) this strongly suggests that the data at least globally represent the *in vivo* assembly kinetics of the corresponding OM TAC module subcomplexes.

Additionally, p197 depletion resulted in an accumulation of the pATOM36- and TAC65-containing subcomplexes (~440 and ~100 kDa) when compared to wild-type cells (Fig 4B). These results suggest that the two classes of detergent-soluble OM TAC module subcomplexes represent distinct assembly intermediates. The observed relative changes in the amounts of the three TAC40-containing subcomplexes (~535, ~770, and ~920 kDa), after p197 and p197/p166 depletion, support a model in which the ~535 kDa TAC40 oligomer assembles first and then sequentially incorporates TAC42 and TAC60 to form the two larger subcomplexes (~770, and ~920 kDa). The accumulation of the largest ~920 kDa TAC40-, TAC42- and TAC60-containing subcomplex, along with the ~440 kDa pATOM36- and TAC65-containing subcomplex, is therefore a direct consequence of the p197 depletion-induced disruption of the cytosolic TAC module, which prevents their integration into the insoluble TAC structure.

## TAC60 overexpression changes the levels of the TAC40-containing subcomplexes

To test the stepwise assembly model outlined in the previous paragraph we used a cell line allowing ectopic tetracycline-dependent expression of TAC60-myc in a wild-type background. The results in Fig 4C shows that after addition of tetracycline, and thus overexpression of TAC60-(myc), the abundance of the ~535 kDa TAC40-oligomer was much reduced, whereas the level of the largest ~920 kDa TAC60-containing subcomplex increased. Moreover, the levels of the ~770

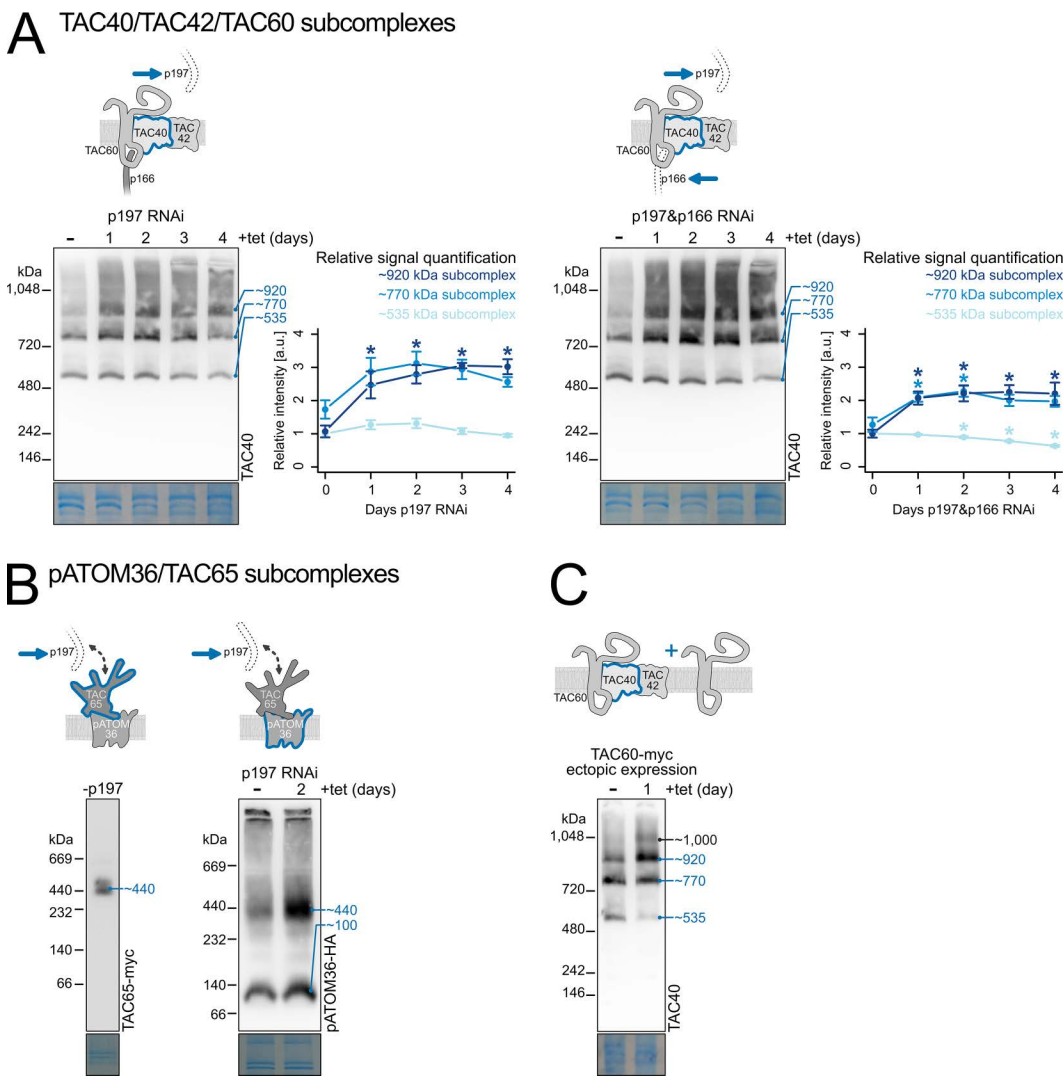

**Fig 4. TAC OM module subcomplexes form independently of cytosolic and inner TAC modules. (A)** Immunoblots of BN-PAGE time course experiments (n = 3 each) using the inducible p197 RNAi cell line (left), or the inducible p197/p166 double RNAi cell line (right) were probed for TAC40. The signal intensity of every subcomplex band (~535 kDa, ~770 kDa, ~920 kDa) was quantified and normalized to the ~535 kDa subcomplex intensity of the sample of the uninduced RNAi cell line for every experiment. ANOVA was performed for data points grouped by subcomplexes, where $P$ value < 0.1 was accepted as marginal significance. Within significant groups, pairwise Student's t-tests were performed and $P$ values were adjusted by the Holm-Bonferroni method. * mark data points with adjusted $P$ values < 0.1 when comparing data points between uninduced (day 0) and induced samples. Error bars indicate the standard error of the mean. For confirmation of the knockdowns see S3 Fig **(B)** as in (A) but immunoblots are from of a cell line expressing TAC65-myc in an induced p197 knockdown (left) and a cell line expressing pATOM36-HA in an inducible p197 knockdown (right) are shown. **(C)** as in (A and B) but immunoblots are from a cell line allowing ectopic tetracycline-dependent expression of TAC60-myc in a wild-type background.

the kDa subcomplex stayed the same and an additional high molecular weight subcomplex of ~1000 kDa, that was not seen in the absence of tetracycline, was also detected. The simplest explanation for these results is that the higher level of TAC60-(myc) caused a more efficient conversion of the ~535 kDa TAC40 oligomer either directly, or more likely via the ~770 kDa subcomplex, into the ~920 kDa subcomplex and even into the yet uncharacterized ~1000 kDa subcomplex. It also suggests that the abundance of TAC60 is a rate-limiting factor for the formation of the ~920 kDa subcomplex.

## Rapid *de novo* formation of TAC40-containing subcomplexes

The *T. brucei* γL262P mutant bloodstream form (BSF), carrying an L262P mutation in the γ -subunit of the ATPase, can grow in the absence of kDNA and does not require the TAC [53]. To investigate TAC assembly, we generated a TAC40 double-knockout γL262P-cell line. While these cells lacked both the OM and inner TAC modules, resulting in the complete loss of kDNA, they exhibited normal growth.

Based on findings in PCFs [35], we expected that other OM TAC module subunits, though delocalized, would still be present within the OM. A limitation of the RNAi analyses shown in Figs 3 and 4 was that small amounts of the targeted proteins, and thus of residual TAC structures that may act as assembly seeds, were still present. However, in the γ L262P TAC40 double-knockout cells, TAC40 and thus the OM and the inner TAC modules were completely absent. Consequently, tetracycline-induced ectopic re-expression of tagged TAC40 in this cell line triggered *de novo* formation of the two modules.

It had previously been shown that TAC subunits are generally expressed at a higher level in BSFs compared to PCF cells [54,55]. Fig 5 shows that wild-type BSFs of *T. brucei* have three TAC40-containing subcomplexes (~535, ~770 and ~920 kDa). Except for the ~920 kDa band, which appears to have a lower relative intensity, this is identical to what was observed in PCFs (Figs 1B and 2A). Thus, BSFs formed the same TAC40-, TAC42- and TAC60-containing subcomplexes than PCFs. Triggering re-expression of tagged TAC40 in the TAC40-lacking γL262P cell line results in a time-dependent *de novo* formation of the three tagged TAC40-containing subcomplexes with a similar pattern than was observed in wild-type BSFs (Fig 5). The migration of these subcomplexes was slightly slower than in wild-type cells, which was likely because they consisted exclusively of tagged TAC40. All three tagged TAC40-containing complexes were already detected after four hours, indicating that their de novo formation was rapid. Moreover, their relative amounts remained the same over time. This is expected because unlike in the experiments shown in Fig 4, p197 was still expressed in the γ L262P cells, which allowed continuous integration of the newly formed ~920 kDa TAC-containing subcomplex into the insoluble TAC structure.

## The TAC60 N-terminus is essential for the integrity of the OM TAC module

TAC60 has two α-helical transmembrane domains and both its N- and C-termini face the cytosol [38]. The short IMS-exposed loop contains the binding site for the C-terminus of p166, the only integral IM TAC subunit of the inner TAC module [40,41]. Previous *in vivo* deletion studies revealed that a tagged TAC60 variant lacking the cytosolic C-terminal 283 aa (TAC60-ΔC283-myc) remained functional. However, a TAC60 variant lacking both the N- and C-terminal domains (TAC60-ΔN97ΔC283-myc) was non-functional despite showing TAC localization [38].

To study the subcomplexes formed by these truncated TAC60 variants, we used previously established RNAi cell lines allowing tetracycline-inducible replacement of full length TAC60 by TAC60-Δ283-myc or TAC60-ΔN97ΔC28-myc, respectively. BN-PAGE analyses showed that TAC60-ΔC283-myc and TAC60-ΔN97ΔC283-myc formed two subcomplexes each (Fig 6A), similar to full-length TAC60. However, as expected, their estimated molecular weights (~735/815 kDa and ~670/750 kDa, respectively) were lower than those formed by full-length TAC60 (~770/920 kDa). Moreover, while the abundance of TAC60-ΔC283-myc is similar in both bands, the band corresponding to the larger subcomplex formed by TAC60-ΔN97ΔC283-myc (750 kDa) was much more intense, suggesting accumulation of a non-productive assembly intermediate.

To confirm these findings, we monitored the morphology of the OM TAC module in the cell line allowing for tetracycline-inducible exclusive expression of TAC60-ΔN97ΔC28-myc. Immunofluorescence analysis of this cell line showed a progressive TAC40 delocalization from newly formed basal bodies, culminating in a complete mislocalization after two days of induction (Fig 6B). The same cells were also analyzed biochemically. In uninduced cells, the majority of TAC40 is integrated into the detergent-resistant insoluble TAC structure recovered in the pellet fractions after a two-step (0.015 and 1%) digitonin fractionation (Fig 6C, P1 and P2). However, in line with the microscopy results, the TAC40 signal shifted from the P2 to the SN2 fraction, indicating that after exclusive expression of TAC60-ΔN97ΔC28-myc much of TAC40 became detergent-soluble. Moreover, also TAC60-ΔN97ΔC28-myc was mostly detergent soluble.

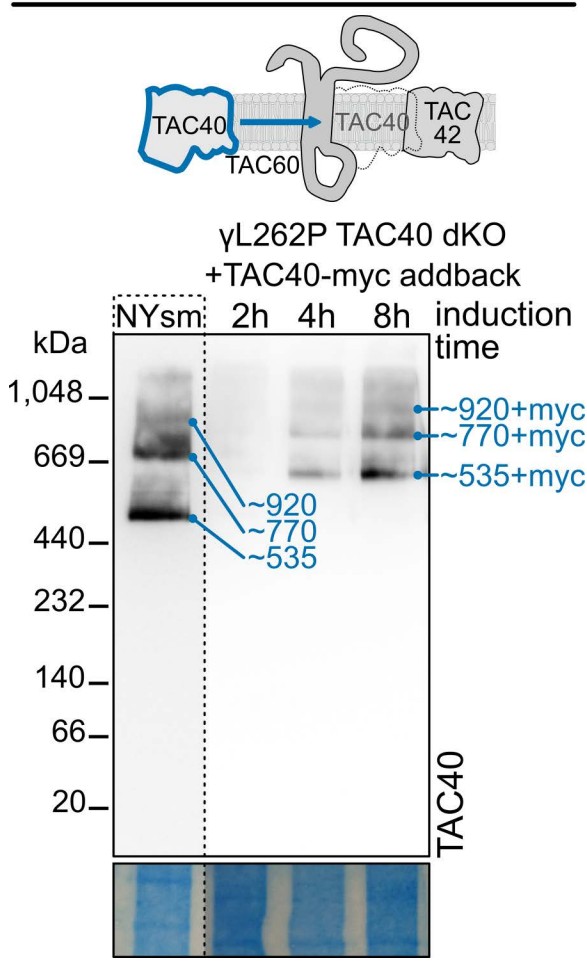

**Fig 5. Rapid de novo formation of TAC40-containing subcomplexes.** Immunoblot of a single BN-PAGE gel from experiments with BSF *T. brucei* cells probed for TAC40. The left lane shows the detergent-soluble TAC40-containing subcomplexes in wildtype NYsm cells. The three lanes on the right show detergent-soluble complexes extracted from a TAC40 dKO γL262P BSF cell line induced for TAC40-myc addback expression for 2, 4, and 8 hours. The scheme (top) visualizes the conditional addback expression of TAC40 and complex formation with TAC42 and TAC60. The NYsm lane was imaged with the less sensitive chemiluminescent substrate "Pico", whereas the three remaining lanes were imaged with the "Femto" substrate detection kit (see Material and Methods). For additional controls of this cell line see S4 Fig.

We conclude that despite forming assembly intermediates with TAC40 and TAC42, TAC60-ΔN97ΔC283-myc failed to integrate into the detergent-insoluble TAC structure, blocking the formation of a functional TAC. Since the C-terminus of TAC60 is dispensable for protein function, these findings demonstrate the essential role of the N-terminus of TAC60 (1–97 aa) for OM TAC module assembly, possibly by facilitating interactions with the pATOM36- and TAC65-containing subcomplex.

## Predicted structure of the TAC40 oligomer

TAC40 independently forms a stable ~535 kDa subcomplex, detectable as a sharp band on BN-PAGE. It also serves as the core subunit of two larger assembly subcomplexes (~770 and ~920 kDa) that incorporate TAC42 and TAC60. Notably,

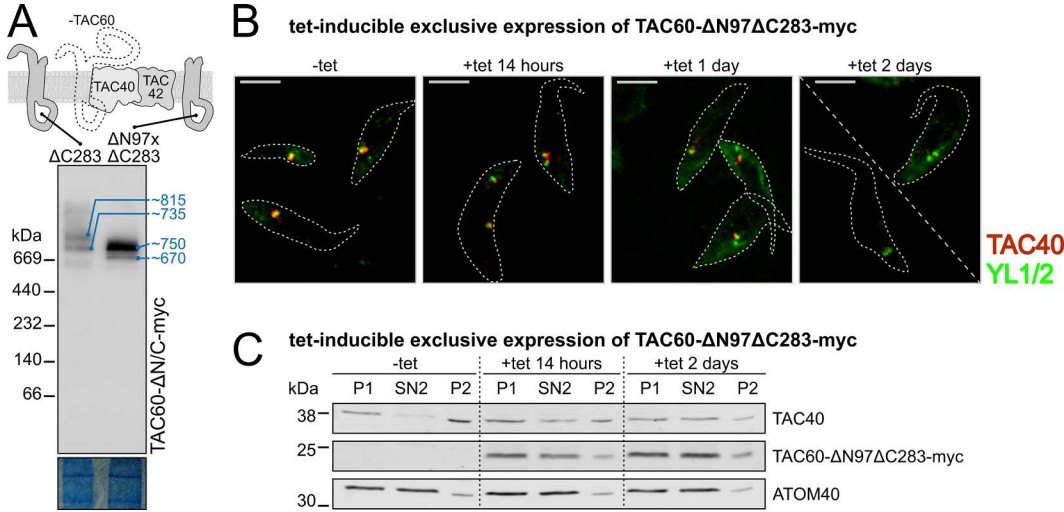

**Fig 6. The TAC60 N-terminus is essential for the integrity of the OM TAC module.** **(A)** Immunoblot of a BN-PAGE experiment of cells expressing the TAC60 truncation variants TAC60-ΔC283-myc and TAC60-ΔN97ΔC283-myc under TAC60 knockdown probed for the TAC60 variants. The scheme at the top depicts the truncated TAC60 variants. **(B)** Immunofluorescence microscopy images of cytoskeletons isolated at indicated time points after tetracycline (tet)-induction of the TAC60-ΔN97ΔC283-myc exclusive expressor cell line. TAC40 (red) and tyrosinated α-tubulin which stains basal bodies (YL1/2, green) were detected using specific antisera. Scale bar: 5 μm. **(C)** Immunoblot of an SDS-PAGE gel of an experiment performed with the identical cell line as in (B) induced for the indicated time with tetracycline. Samples of the organellar fraction (P1), the soluble organellar fraction (SN2) and the insoluble organellar fraction (P2) were collected as described in the Material and methods. TAC40 and TAC60-ΔN97ΔC283-myc were detected using a anti TAC40 antiserum and a myc specific antibody, respectively. ATOM40 serves as a marker for the soluble organellar fraction.

TAC42 and TAC60 cannot form detectable subcomplexes on their own and instead must assemble onto the preexisting TAC40-only complex.

The existence of the TAC40 oligomer is strongly supported by the results shown in Figs 1 and 2. Moreover, BN-PAGE analysis of an anti-HA pulldown experiment, performed on extracts from a TAC42-depleted cell line expressing both untagged and HA-tagged TAC40 alleles, exclusively recovered the ~535 kDa TAC40 oligomer under native elution conditions (Fig 7A). Finally, when analyzed by denaturing SDS-PAGE both HA-tagged and wild-type TAC40 are detected, confirming that TAC40 molecules interact with each other within this complex (Fig 7B).

To determine the number and arrangement of TAC40 molecules within the ~535 kDa oligomer, we performed an *in silico* analysis using AlphaFold3 [56]. The predicted template modeling score (pTM) for a single TAC40 molecule was 0.79, indicating a highly accurate prediction of its structure (Fig 7C). Since AlphaFold3 can also predict protein complexes, we modeled TAC40 oligomers containing 1–9 TAC40 molecules (Fig 7D). In the resulting predictions, TAC40 molecules consistently formed symmetrical ring-like structures. In all cases, the proteins maintained a consistent topology in a planar arrangement, in line with their integral membrane localization.

The most confident structural prediction was obtained for the TAC40 heptamer, which had a pTM score of 0.64 and an interface pTM (ipTM) score of 0.61 (Fig 7E). The hexamer and octamer showed slightly lower scores, while other oligomeric states were predicted with low confidence only.

## Discussion

The fully assembled TAC is a very large, permanent structure that is insoluble in non-ionic detergents [32,41,42]. However, subunits of the OM TAC module are not only present in the final TAC structure but are also recovered in two groups of detergent-soluble subcomplexes. The first group consists of three subcomplexes of ~535, ~770, and ~920 kDa that

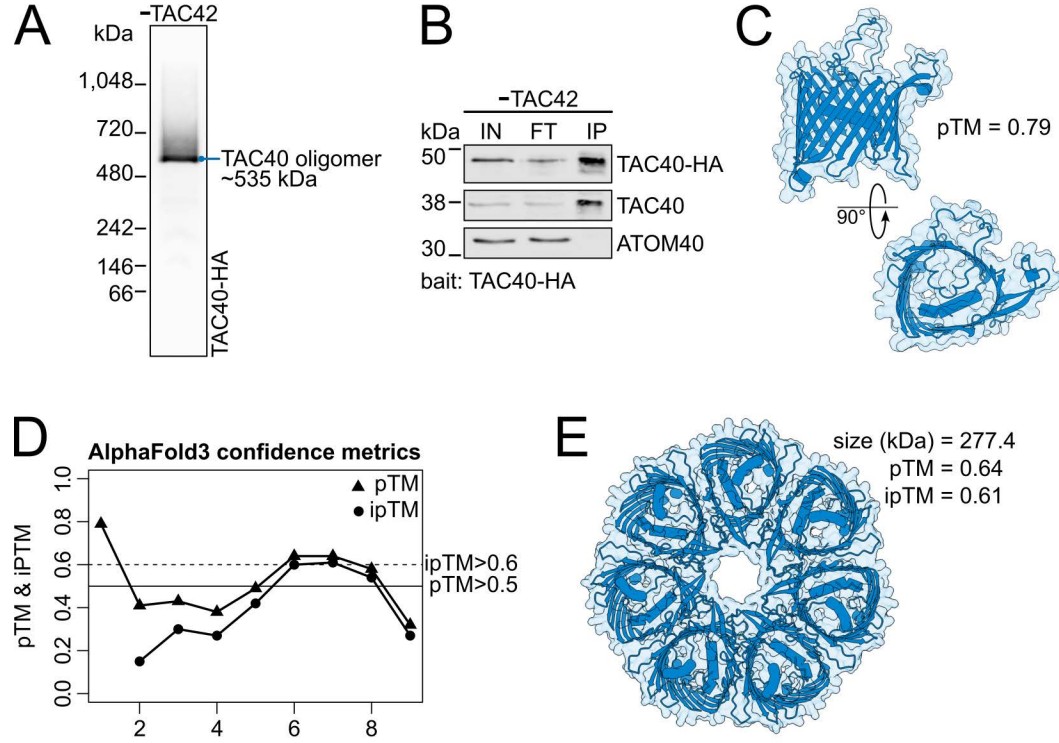

**Fig 7. Predicted structure of the TAC40 oligomer. (A)** BN-PAGE immunoblot probed for TAC40-HA of the TAC40-HA oligomer-containing eluate of an HA-pulldown experiment. The TAC40-HA oligomer was pulled down from the extract of a cell line depleted for TAC42 that expressed one HA-tagged TAC40 allele. **(B)** Fractions of the same HA-pulldown experiment shown in (A) were analyzed by an SDS-PAGE immunoblot and probed for TAC40-HA, TAC40, and ATOM40. IN: input (digitonin-solubilized mitochondria enriched fraction) $5 \times 10^6$ cell equivalents; FT: Flow through, $5 \times 10^6$ cell equivalents; IP: eluate, $1 \times 10^8$ cell equivalents. ATOM40 serves as a negative control. **(C)** AlphaFold3 predicted structure model of the TAC40 monomer. The model of the predicted structure is shown from the side (top) and from IMS side (bottom). pTM: predicted template modelling score. **(D)** Graph showing the pTM and the interference pTM (ipTM) scores of AlphaFold3 predicted structures of complexes formed by TAC40 containing 1-9 monomers. Threshold lines for the pTM (>0.5) and ipTM (>0.6) are indicated with a solid and dashed line, respectively. **(E)** Top view of an Alphafold3 predicted structure model of the heptameric TAC40 complex.

contain variable amounts of TAC40, TAC42 and TAC60, whereas the second group consists of a ~440 kDa subcomplex formed by pATOM36 and TAC65 (Figs 1 and 2).

AlphaFold3 predicts with high confidence (pTM/ipTM scores > 0.6) that the ~535 kDa subcomplex of the first group consists of an oligomer containing 6–8 molecules of TAC40 that are arranged in a planar ring (Fig 7D, 7E). This prediction is consistent with the BN-PAGE and immunoblots analyses which showed a sharp band of ~535 kDa that contained TAC40 but neither TAC42 nor TAC60. Moreover, it also fits with the known integral membrane localization of TAC40 and could explain how it can assembly into such a well-defined subcomplex. However, the molecular weight of the TAC40 subcomplex of ~535 kDa, as determined by BN-PAGE, is much higher than the calculated 240–320 kDa for the TAC40 hexa- to octamer. A possible explanation for this discrepancy is that the TAC40 oligomer forms a planar ring with a central cavity rather than a globular complex, leading to aberrant migration on BN-PAGE. Additionally, the TAC oligomer is membrane-embedded and solubilized by detergents which likely causes a shift towards a higher molecular weight on a BN-PAGE.

TAC40 is a kinetoplastid-specific β-barrel protein that belongs to the VDAC-like protein family [39]. It has been reported that mammalian VDAC is present in a dynamic equilibrium between dimers and oligomers [57,58]. The function of these VDAC oligomers appear to be connected to apoptosis-related cytochrome c release from the IMS [58]. Moreover, in

oxidatively stressed mitochondria VDAC1 oligomers have been implicated in the release of short mtDNA fragments to the cytosol where they cause inflammation [57]. It is therefore not surprising that TAC40 also appears to oligomerize. However, whereas mammalian VDAC oligomers are stress-induced and form large pores in the OM, the trypanosomal TAC40 is present exclusively in the oligomeric form and its function is restricted to the organization of the TAC architecture, which to our knowledge does not require pore formation. This is evidenced by the fact that TAC40 is dispensable in the γL262P cell line, that can grow in the absence of kDNA [53] (Figs 5 and S5).

Seven lines of evidence suggest that the three TAC40-containing subcomplexes (~535, ~770, ~920 kDa), as well as the subcomplex consisting of pATOM36 and TAC65 (~440 kDa), are assembly intermediates of the OM TAC module:

(i) Unlike in the fully assembled TAC, the OM TAC module subunits are detergent-soluble when present in the four subcomplexes (Fig 1).

(ii) The smallest subcomplex (~535 kDa) has the simplest composition consisting of TAC40-only, whereas the two larger subcomplexes (~770, ~920 kDa) in addition to TAC40 also contain TAC42 and TAC60 (Fig 2A).

(iii) Neither TAC42 nor TAC60 form subcomplexes on their own, or with each other, suggesting that TAC42 and TAC60 are incorporated into the preexisting TAC40-only subcomplex (Fig 2A).

(iv) The four subcomplexes (~535, ~770, ~920 and ~440 kDa) form independently of the cytosolic and the inner TAC modules (Fig 4A, 4B).

(v) Depletion of p197 results in the accumulation of the two higher TAC40-containing subcomplexes (~770, ~920 kDa), likely because the 920 kDa subcomplex cannot be properly linked to the nascent TAC structure. The pATOM36- and TAC65-containing subcomplex (~440 kDa) likely accumulates for the same reason (Fig 4A, 4B).

(vi) Overexpression of TAC60 shifts the steady state levels of the three subcomplexes: the ~535 kDa TAC40 oligomer gets depleted whereas the ~920 kDa TAC40/TAC42/TAC60-containing subcomplex accumulates (Fig 4C).

(vii) Deleting the N-terminal 97 aa of TAC60, which prevent its incorporation into the nascent TAC structure [38] - similar to what was observed after depletion of p197 (Fig 4) - causes a massive accumulation of the largest TAC40-, TAC42- and TAC60-containing subcomplex (corresponding to 920 kDa in wild-type cells) (Fig 6A).

Based on these results we propose the following working model for the assembly pathway of the OM TAC module.

In the first step, TAC40 is inserted into the OM mediated by the ATOM complex and by the β-barrel protein insertion pore Sam50 [39,59]. TAC40 then oligomerizes into a ring-like structure likely composed of 6–8 molecules, forming a platform that provides binding sites for TAC42 and TAC60 (Fig 8A).

In the second step, TAC42 and TAC60 associate with the preformed TAC40 oligomer in two stages, forming first the ~770 and subsequently the ~920 kDa subcomplexes (Fig 8B). The two proteins may be incorporated individually or as a dimer. The existence of a TAC42-TAC60 dimer is predicted by AlphaFold3 with good confidence (pTM = 0.42, ipTM = 0.68) (S5A Fig). Moreover, consistent with our working model, AlphaFold3 predictions using seven TAC40 molecules and one or two TAC42-TAC60 pairs as templates generate structures that potentially correspond to two subcomplexes with pTM/ipTM scores > 0.36 (S5B, S5C Fig). These structures appear plausible, as the two transmembrane domains of TAC60 align within the OM plane with the correct topology ($N_{Cyotoslic}$, $C_{Cytosolic}$) relative to the β-barrel proteins TAC40 ($N_{IMS}$, $C_{IMS}$) and TAC42 ($N_{IMS}$, $C_{IMS}$).

Much of the third step remains speculative. Since TAC65 interacts with p197 [32], the ~440 kDa subcomplex containing pATOM36 and TAC65 should be able to interact with the cytosolic TAC module subunit p197 on its own. However, TAC65 fails to assemble into the TAC after depletion of either TAC40 or TAC60 [35]. While the molecular mechanism underlying the crosstalk between the two subcomplex classes remains unclear the N-terminal 97 aa of TAC60 are essential for the

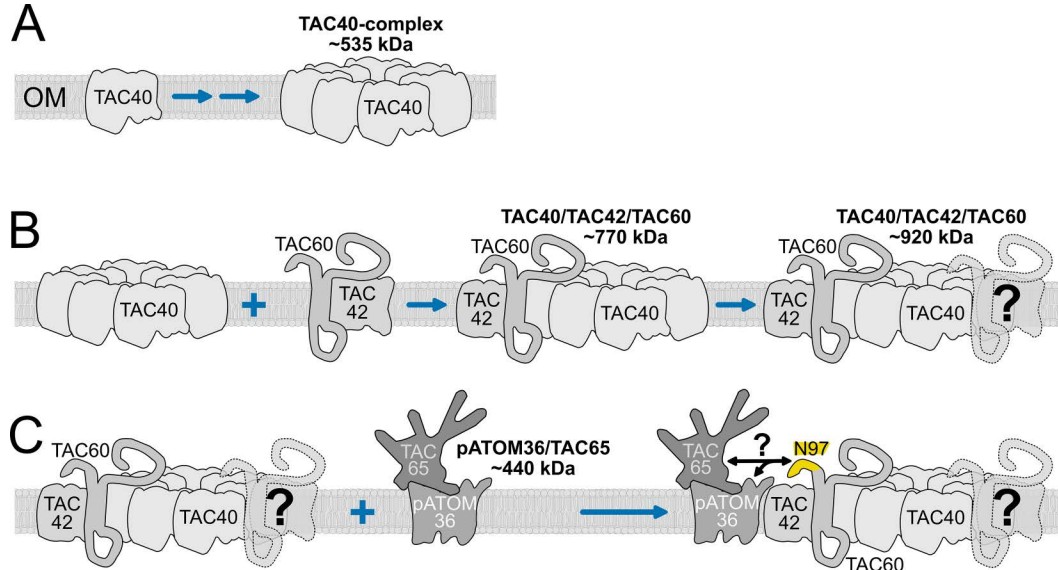

**Fig 8. Assembly model of the OM TAC module.** (A) Oligomerization of TAC40 into a ~535 kDa TAC40 subcomplex. OM: mitochondrial outer membrane. **(B)** The TAC40 oligomer binds to either a dimer or monomers of TAC42 and TAC60 forming the ~770 kDa and subsequently the ~920 kDa TAC40/TAC42/TAC60 subcomplex. **(C)** The ~920 kDa TAC40/TAC42/TAC60 subcomplex interacts with pATOM36/TAC65 subcomplex (~440 kDa) resulting in a putative OM TAC modules assembly intermediate.

formation of the TAC OM module (Fig 6B, 6C). This suggests that merging of the ~920 kDa subcomplex (TAC40, TAC42, TAC60) with the ~440 kDa complex (pATOM36, TAC65) (Fig 8C) is a prerequisite for linking the OM TAC module to p197 and, consequently, to the cytosolic TAC module.

After its integration in the nascent TAC structure the OM TAC module interacts with the matrix-localized p166 via TAC60. If the fully formed OM TAC module is considered as a single unit, formation of the TAC follows the hierarchical assembly model, which implies a strict sequential assembly of TAC subunits from the (pro)basal body to the kDNA [35]. However, formation of the OM TAC module itself is independent of the cytosolic and the inner TAC modules. It follows a unique pathway involving at least four membrane-embedded subcomplexes. The first one consists of the TAC40 oligomer, two further subcomplexes are formed by adding various amounts of TAC42 and TAC60 to the oligomer. Subsequently, the largest of TAC40-containing subcomplex merges with the separately formed pATOM36- and TAC65-containing subcomplex forming the fully assembled OM TAC module.

It is not surprising that the assembly of the OM TAC module is so complicated, since with five different subunits it is the most intricate of all three TAC modules. Moreover, except for TAC65, all of its subunits are integral membrane proteins that need to be inserted into the OM, a process mediated by at least two different insertases: the ATOM complex and Sam50 [14].

It is important to acknowledge the limitations of our assembly model. While the composition of various assembly subcomplexes is known, the stoichiometry of their subunits remains to be determined. The proposed TAC40 oligomer structure is predicted by AlphaFold with high confidence and supported by experimental evidence. However, the same confidence does not extend to the predicted structures of the other subcomplexes. A deeper understanding of the OM TAC module assembly pathway will require experimental determination of the atomic structure of its subcomplexes, which is beyond the scope of this study. Despite these limitations, our model provides a valuable framework that can guide

future experiments aimed at elucidating the assembly pathway of the TAC structure, the unique mitochondrial genome segregation system of trypanosomes.

## Materials and methods

### Soluble fraction Co-IP protein-protein interaction network

Protein-protein interaction networks were constructed in Cytoscape (version 3.9) [60]. Relative mean fold change values (mean fold change/mean fold change of bait) irrelevant of the corresponding $P$ values were calculated for all proteins with a mean fold change > 1.0 in any of the previously published TAC40, TAC4, TAC60, TAC65 and pATOM36 pulldown experiments [36,38]. Possible bait-bait interactions were ignored for network construction due to technical reasons. The relative fold change was used as a metric to construct edge-weighted spring embedded layouts.

### Transgenic cell lines

Procyclic cell lines are based on the *T. brucei* 29–13 [61] and a single marker *T. brucei* 427 strain [32] and were grown at 27°C in SDM-79 supplemented with 10% or 5% (v/v) fetal calf serum, respectively. BSF cell lines are based on the NYsm strain [61] or on the γL262P variant [53] and were cultivated in HMI-9 containing 10% (v/v) fetal calf serum at 37°C in a 5% (v/v) $CO_2$ atmosphere. RNAi cell lines of the TAC proteins used in the study were created using modified pLEW100 vectors [61] which contain stem loops allowing the expression of double stranded RNAs corresponding to open reading frames (ORF) or to the 3' untranslated regions (UTR) of the target mRNAs, respectively. An overview of all used cell lines is given in Table 1. The p166/p197 double RNAi cell line was produced by the stable transfection of a p197 3'UTR RNAi cell line [32] with a p166 3'UTR RNAi stem loop vector [41].

C-terminal *in situ* 3xHA tags for TAC40, TAC42, and pATOM36 were introduced using PCR products amplified from vectors of the pMOtag series [62]. Transfection of these PCR products allowed the tagging of one of the endogenous alleles (Table 1).

C-terminally tagged variants of TAC65 as well as the wild type TAC60 and truncated variants thereof were expressed under tetracycline control from ectopic genes. The constructs used for stable transfection are based on modified pLEW100 vectors [61] and have been used before (Table 1).

To generate a double allele knockout of TAC40 in the γL262P cell line, 500 base pairs of 5' and 3' flanking regions of TAC40 were cloned upstream and downstream of the resistance cassettes of vectors of the pMOtag series [62]. This way, a blasticidine and a phleomycin resistance gene were used to produce a single and double allele knockout of TAC40, respectively. For inducible addback expression, we used a modified pLEW100 vector [61] allowing the stable integration into the rDNA locus of a TAC40 gene with a C-terminal 3x myc tag whose expression is regulated by tetracycline [39].

### Cell fractionation

One or two step digitonin extractions were used to isolate mitochondria-enriched fractions and to prepare solubilized mitochondrial extracts, respectively [32]. Unless stated otherwise, cell lines with knocked in constructs for the ectopic expression of tagged proteins of interest or RNAi constructs targeting the gene of interest have been induced with tetracycline (1.0 µg/ml) for 2 days. To study the subcellular localization of TAC65 (Fig 2C), $5 \times 10^7$ cells were collected and washed twice in PBS (137 mM NaCl, 2.7 mM KCl, 10 mM $Na_2HPO_4$, and 1.8 mM $KH_2PO_4$, pH 7.4). After resuspension in 0.25 ml of SoTE buffer (20 mM Tris HCl pH 7.5, 0.6 M sorbitol, 2 mM ethylenediaminetetraacetic acid (EDTA), 1x cOmplete, Mini, EDTA-free protease-inhibitor-cocktail (Roche)) 0.25 ml of SoTE containing 0.03% (w/v) digitonin was added at room temperature. After a 10 min incubation on ice, a mitochondria-enriched pellet (P1) was separated from the cytosolic fraction (SN1) by centrifugation (6'700 g, 5 minutes, 4°C).

For BN-PAGE analysis the P1 fraction of a digitonin extraction corresponding to $10^8$ cell equivalents was resuspended in 100 µl 20 mM Tris HCl pH 7.4, 100 mM NaCl, 10% glycerol, 0.1 mM EDTA, 1% (w/v) digitonin and 1x of cOmplete

**Table 1. Summary of transgenic *T. brucei* cell lines used in this study.**

| Description | Parental cell line | Additional information | Reference | Figure |
|---|---|---|---|---|
| TAC42 3x HA | | C-term 3x HA (*in situ*, single allele) | Cell line: [38] | Fig 1B middle |
| TAC60 3x myc | | C-term 3x myc (ectopic allele) | Cell line: [38] | Figs 1B right, 2A right, 4C |
| TAC65 3x myc | | C-term 3x myc (ectopic allele) | Cell line: [36] | Figs 1C left, 2B left, 2C left. |
| pATOM36 3x HA | | C-term 3x HA (*in situ*, single allele) | Cell line: [36] | Fig 1C right |
| TAC60 ORF RNAi | | | Cell line: [38] | Fig 2A left |
| TAC42 ORF RNAi | | | Cell line: [38] | Fig 2A left |
| TAC40 ORF RNAi | | | Cell line: [39] | S2 Fig. |
| TAC42 3x HA x TAC60 ORF RNAi. | TAC60 ORF RNAi [38] | C-term 3x HA (*in situ*, single allele) (PCR based, [38]) | This study | Fig 2A middle |
| TAC42 3x HA x TAC40 ORF RNAi | TAC40 ORF RNAi [39] | C-term 3x HA (*in situ*, single allele) (PCR based, [38]) | This study | Fig 2A middle |
| TAC60 3x myc x TAC42 ORF RNAi | TAC42 ORF RNAi [38] | C-term 3x myc (ectopic allele) [38] | This study | Fig 2A right |
| TAC60 3x myc x TAC40 ORF RNAi | TAC40 ORF RNAi [39] | C-term 3x myc (ectopic allele) [38] | This study | Fig 2A right |
| TAC65 3x myc x pATOM36 ORF RNAi | pATOM36 ORF RNAi [52] | C-term 3x myc (ectopic allele) [36] | This study | Fig 2B left, 2C right |
| pATOM36 3x HA x TAC65 ORF RNAi | TAC65 ORF RNAi [36] | C-term 3x HA (*in situ*, single allele) (PCR based, [52]) | This study | Fig 2B right |
| pATOM36 ORF RNAi | | | Cell line: [52] | Fig 3A left |
| TAC65 ORF RNAi | | | Cell line: [36] | Fig 3A right |
| TAC65 3x myc x TAC40 ORF RNAi | TAC40 ORF RNAi [39] | C-term 3x myc (ectopic allele) [36] | This study | Fig 3B left |
| pATOM36 3x HA x TAC40 ORF RNAi | TAC40 ORF RNAi [39] | C-term 3x HA (*in situ*, single allele) (PCR based, [52]) | This study | Fig 3B right |
| p197 3'UTR RNAi | | | Cell line: [32] | Fig 4A left |
| p197 3'UTR RNAi x p166 3'UTR RNAi | p197 3'UTR RNAi [32] | 2nd RNAi: p166 3'UTR RNAi (Vector: [41]) | This study | Fig 4A right |
| TAC65 3x myc x p197 3'UTR RNAi | p197 3'UTR RNAi [32] | C-term 3x myc (ectopic allele) [36] | This study | Fig 4B left |
| pATOM36 3x HA x p197 3'UTR RNAi | p197 3'UTR RNAi [32] | C-term 3x HA (*in situ*, single allele) (PCR based, [52]) | This study | Fig 4B right |
| Bloodstream form NYsm | | | Cell line: [61] | Fig 5 |
| γL262P mutant bloodstream form | | | Cell line: [53] | S4 Fig. |
| Bloodstream form TAC40 dKO x TAC40 3x myc. | γL262P mutant bloodstream form [53] | C-term 3x myc (ectopic allele) [38] | This study | Fig 5, S4 Fig |
| TAC60-ΔC283 3x myc x TAC60 ORF RNAi | | | Cell line: [38] | Fig 6A |
| TAC60-ΔN97ΔC283 3x myc x TAC60 ORF RNAi | | | Cell line: [38] | Fig 6A, 6B, 6C |
| TAC40 3x HA x TAC42 ORF RNAi | TAC42 ORF RNAi [38] | C-term 3x HA (*in situ*, single allele) (PCR based, [39]) | This study | Fig 7A, 7B |

protease-inhibitor-cocktail. The sample was kept on ice for 15 minutes before the final centrifugation (20'000 g, 15 minutes, 4°C) resulting in the SN2 fraction containing solubilized mitochondria and the P2 pellet.

## BN-PAGE

BN-PAGE was used to study native protein complexes. 90 µl of the SN2 supernatant of the two-step digitonin extraction (see above) was mixed with 10 µl 10x loading dye (300 µM Coomassie brilliant blue G-250 (Sigma), 500 mM 6-amino n-caproic acid, 100 mM Bis-Tris pH 7.0) and protein complexes were separated on 4–13% (or 4–10% in Fig 4A and 4C)

polyacrylamide gradient gels. Gradient gels were produced with a peristaltic pump coupled to a two-chambered gradient mixer filled with 4% acrylamide buffer (ratio acrylamide/bisacrylamide = 16.5/1, 67 mM 6-amino n-caproic aid, 50 mM Bis-Tris pH 7.0) and 13% (or 10%) acrylamide buffer (ratio acrylamide/bisacrylamide = 16.5/1, 67 mM 6-amino n-caproic aid, 50 mM BisTris pH 7.0, 17.5% glycerol). The 4% acrylamide buffer was used to pour sample wells adding 3–5 mm pseudo stacking gel with continuous 4% polyacrylamide on top of the gradient gels. Gel electrophoresis was performed with a two buffer systems with a single anode buffer (50 mM BisTris pH 7.0) and two cathode buffers (Cathode buffer A: 50 mM tricine, 15 mM BisTris, 0.12 µM Coomassie brilliant blue G-250, pH 7.0, Cathode buffer B: 50 mM tricine, 15 mM BisTris, pH 7.0) where cathode buffer A was used for migration through the first half of the gel and cathode buffer B for the second half. Electrophoresis was performed at a maximal voltage of 100 V and maximal current of 15 mA (4–5 hours run-time). For immunoblotting. gels were incubated for 5 minutes in 25 mM Tris, 190 mM glycine, 1 mM EDTA, 0.05% sodium dodecyl sulfate (SDS) and subsequently electrophoretically transferred in 20 mM Tris, 150 mM glycine, 0.02% SDS, 20% methanol onto polyvinylidene fluoride membranes (Immobilon-FL).

Proteins of interest were detected by protein-specific or tag-specific primary antibodies followed by horseradish peroxidase (HRP)-coupled secondary antibodies (see below). Finally, the SuperSignal West Pico Plus and Femto chemiluminescent substrate detection kits (Thermo Fisher Scientific) were used for image acquisition.

## Subcomplex size estimation on BN-PAGE gels

Migration distances of marker proteins on BN-PAGE gels used in this study revealed a very strong correlation (Pearson's r > 0.99) between the relative migration distances (defined as distance between the upper edge of the gel and the band of interest, relative to the distance between the upper edge of the gel and the lowest molecular weight marker) and the square root (sqrt) of the protein weight in kDa (S6 Fig). The estimated molecular weights of the TAC subcomplexes presented here were calculated by linear models based on marker protein migration patterns. All models and their coefficients of determination are shown in S6 Fig.

## RNA extraction and reverse transcription PCR

To determine RNAi efficiency in the p166/p197 double RNAi cell line, total RNA of uninduced and two days tetracycline-induced cells was extracted using guanidinium thiocyanate-phenol-chloroform and dissolved in milli-Q water as described in [63]. Total RNA extracts were first treated with DNase to remove the genomic DNA (DNA-free Kit, Ambion). For the reverse transcription, first-strand cDNA synthesis was performed using oligo(dT)20 primers (SuperScript First Strand, Invitrogen). Control reactions without reverse transcriptase were performed simultaneously. Quantitative PCR was done with identical amounts of each cDNA sample and with primer pairs for the amplification of a α-tubulin segment (ORF nt 546-1'249) using primers described in [32], the p197 3'UTR, primer as in [32] and a p166 segment (ORF 3'601–4'189; forward: CAGAAAGCGGTAGAGCACTTTGC; reverse: GCACAGGCGACAATACTTGAACC). PCR products were separated on a 1% agarose gel and stained with ethidium bromide.

## Immunofluorescence microscopy

$10^6$ cells of an exponentially growing cell culture were harvested by centrifugation (2'700 g, 1 minute, room temperature) and washed with PBS. The cells were resuspended in 50 µl PBS and distributed on glass slides where they were allowed to settle before lysis for 30 seconds using PBS containing 0.2% Triton X-100. Following cell lysis, the samples were washed with PBS and fixed in 4% paraformaldehyde for 10 minutes. Fixed samples were washed with PBS and blocked with PBS containing 2% (w/v) bovine serum albumin (BSA) before incubation with two rounds of the corresponding primary and secondary antibodies diluted in PBS containing 2% BSA. For more information on the antibodies used see below. After antibody incubation, slides were washed with PBS, air-dried, and mounted with Vectashield containing 4',6-diamidine-2-phenylindole dihydrochloride (DAPI) (Vectorlabs). The slides were imaged on a DMI6000B microscope

equipped with a DFC360 FX monochrome camera and LAS X software (Leica Microsystems). Images were processed using Fiji software.

## Immunoprecipitations

For immunoprecipitation purification of TAC40-HA complexes, $3 \times 10^8$ exponentially growing cells were harvested and washed twice with PBS. Cells were subjected to a two-step digitonin cell fractionation as described above. The SN2 fraction was incubated with an anti-HA affinity matrix (Roche) for 2 hours at 4°C. Beads were washed five times with a wash buffer (20 mM Tris HCl pH 7.4, 100 mM NaCl, 10% glycerol, 0.1 mM EDTA, and 0.1% (w/v) digitonin). Bound proteins and protein complexes were eluted under native conditions with elution buffer (20 mM Tris HCl pH 7.4, 100 mM NaCl, 25 mM KCl, and 0.1 mM EDTA containing 0.25% (w/v) digitonin and 1mg/ml HA peptide (Sigma)) at 30°C for 15 minutes and further analysed by BN-PAGE. Alternatively, elution was done under denaturing conditions with SDS-PAGE sample buffer without β-mercaptoethanol for SDS-PAGE analysis.

## Antibodies

Dilutions used for immunoblot (IB) and immunofluorescence (IF) analyses are indicated in brackets. The polyclonal rabbit antisera against TAC40 (IB 1:100, IF 1:50) and ATOM40 (IB 1:10'000) were described before [32,64]. The monoclonal rat anti-YL1/2 antibody (IF 1:1'000) that recognizes tyrosinated α-tubulin [65] and the basal body protein TbRP2 [66] was a kind gift from Keith Gull. Commercially available antibodies were used as follows: monoclonal mouse anti-myc antibody (Invitrogen, 132500; IB 1:2'000), monoclonal mouse anti-HA antibody (Sigma, H9658; IB 1:5'000), monoclonal mouse anti-eEF1α antibody (Merck Millipore, 05–235; WB 1:10'000).

Secondary antibodies used for SDS-PAGE immunoblot analyses were IRDye 680LT goat anti-mouse (LI-COR Biosciences, 926–68020; IB 1:20,000) and IRDye 800CW goat anti-rabbit (LI-COR Biosciences, 926–32211; IB 1:20'000), and secondary antibodies used for BN-PAGE immunoblot analyses were HRP-coupled goat anti-mouse antibodies (Sigma, 12–349; IB 1:5'000) as well as goat anti-rabbit antibodies (Sigma, AP307P; IB 1:5'000). Secondary antibodies used for immunofluorescence analyses were goat anti-rat Alexa Fluor 488 (Thermo Fisher Scientific, A-Z1006; IF 1:1'000), goat anti-rabbit Alexa Fluor 596 (Thermo Fisher Scientific, A-11012; IF 1:1'000).

## Supporting information

**S1 Fig. Molecular model of the tripartite attachment complex (TAC).** On the molecular level the TAC can be divided into three modules: (i) the cytosolic TAC module, (ii) the OM TAC module, and (iii) the inner TAC module. The cytosolic TAC module consists exclusively of p197, a protein anchored at the basal body which connects to an unknown domain of TAC65 at the OM. The OM TAC module contains the peripheral membrane protein (TAC65), two integral membrane proteins (pATOM36, TAC60) with α-helical transmembrane domains, and two β-barrel membrane proteins (TAC40, TAC42). TAC65 and pATOM36 interact, the same is the case for TAC40, TAC42, and TAC60. How the two groups of proteins interact with each other is unclear (?). TAC60 interacts with p166, an integral IM protein with a single α-helical transmembrane domain. As a part of the inner TAC module, p166 interacts with TAC102 in the mitochondrial matrix. TAC53 is the inner TAC module subunit that is most proximal to the kDNA.
(TIF)

**S2 Fig. Control of TAC40 antisera on BN-PAGE separated procyclic *T. brucei* samples.** Immunoblot of a BN-PAGE experiment with an uninduced (-tet) and TAC40 RNAi induced (+tet 2 days) procyclic *T. brucei* cell line probed with the TAC40 antibody. The positions of marker proteins with their size in kDa are indicated on the left. Coomassie blue-stained gel sections (bottom) serve as loading controls.
(TIF)

**S3 Fig. Controls for the inducible p166/p197 double RNAi cell line.** Ethidium bromide-stained agarose gels of PCR amplified cDNA segments corresponding to p166, p197, and tubulin mRNAs, as well as non-amplified cytosolic rRNA of the uninduced and induced p166/p197 double RNAi cell line. Tubulin cDNA and rRNA signals serve as loading controls. (TIF)

**S4 Fig. Controls for the TAC40 dKO γL262P BSF cell line allowing inducible ectopic expression of TAC40-myc.** Immunoblots comparing whole cell protein levels of TAC40-myc (top panel), TAC40 (middle panel), and ATOM40 (bottom panel). The parent γL262P BSF cell line (left lane) and TAC40 dKO γL262P BSF uninduced (middle lane) and induced (right lane) for ectopic expression of TAC40-myc were analyzed. The ATOM40 signals serve as a loading control. Numbers on the left indicate protein size markers in kDa. Asterisk, unspecific band recognized by the polyclonal TAC40 antiserum. (TIF)

**S5 Fig. AlphaFold3 structure predictions for the TAC40/TAC42/TAC60 subcomplexes. (A)** Model of an AlphaFold3 structure prediction for a TAC42/TAC60 dimer shown from the side of the membrane. TMD: α-helical transmembrane domain; pTM: predicted template modelling score; ipTM: interference pTM. **(B)** Model of an AlphaFold3 structure prediction for a complex possibly matching the ~770 kDa TAC40/TAC42/TAC60 subcomplex shown from the side of the membrane. **(C)** Model of an AlphaFold3 structure predictions for a complex possibly matching the ~920 kDa TAC40/TAC42/TAC60 subcomplex shown from the cytosolic (left) and the inter-membrane space (right) side. (TIF)

**S6 Fig. BN-PAGE complex size estimation and extrapolation. (A)** Definition of "relative migration distance": distance between the upper edge of the gel and the band of interest divided by the distance between the upper edge of the gel and the lowest molecular weight marker. The shown lane is identical to the panel in Fig 1B (left). **(B-F)** Graphs depicting the linear models of the relative migration distance of marker proteins versus the square root (sqrt) of the molecular weight in kDa. Dark gray datapoints represent marker protein data used for the linear models. Light gray datapoints in (E) and (F) show marker measurements which were omitted for calculation of linear models. The solid and dashed gray lines show the inter- and extrapolated linear models, respectively. Dotted lines show the 90% prediction confidence intervals. The coefficient of determination ($R^2$) of the linear model is shown at the bottom left of each graph. (TIF)

**S1 Data. Numerical data for all graphs presented in the study.**
(XLSX)

**S1 Raw images. Full scans of gels and blots.**
(PDF)

## Author contributions

**Conceptualization:** Philip Stettler, André Schneider.

**Formal analysis:** Philip Stettler, Salome Aeschlimann, André Schneider.

**Investigation:** Philip Stettler, Salome Aeschlimann, Bernd Schimanski.

**Resources:** André Schneider.

**Supervision:** Bernd Schimanski, André Schneider.

**Validation:** Philip Stettler.

**Visualization:** Philip Stettler.

**Writing – original draft:** Philip Stettler, André Schneider.

**Writing – review & editing:** Philip Stettler, Salome Aeschlimann, Bernd Schimanski, André Schneider.

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
