## [Decision Letter · Decision Letter 0]

29 Jun 2025

Assembly of the mitochondrial outer membrane module of the trypanosomal tripartite attachment complex

PLOS Pathogens

Dear Dr. Schneider,

Thank you for submitting your manuscript to PLOS Pathogens. After careful consideration, we feel that it has merit but does not fully meet PLOS Pathogens's publication criteria as it currently stands. Therefore, we invite you to submit a revised version of the manuscript that addresses the points raised during the review process.

Please submit your revised manuscript within 60 days Aug 28 2025 11:59PM. If you will need more time than this to complete your revisions, please reply to this message or contact the journal office at plospathogens@plos.org. Please include the following items when submitting your revised manuscript:

We look forward to receiving your revised manuscript.

Kind regards,

Dominique Soldati-Favre

Section Editor

PLOS Pathogens

Editor-in-Chief

PLOS Pathogens

orcid.org/0000-0003-2946-9497

Editor-in-Chief

PLOS Pathogens

orcid.org/0000-0002-7699-2064

**Additional Editor Comments :**

Both reviewers agree that the manuscript places excessive reliance on BN-PAGE data. They recommend incorporating additional, complementary experimental approaches to reinforce the conclusions and more precisely define the functional roles of the TAC subcomplexes. In addition, the reviewers underscore the importance of ensuring consistent and meaningful sample comparisons, specifically by co-expressing tagged proteins within the same cell line, a concern that could be addressed either through a clear rationale or experimentally. They also stress the need for appropriate controls to definitively rule out nonspecific antibody signals in the presented analyses.

**Journal Requirements:**

At this stage, the following Authors/Authors require contributions: Philip Stettler, Salome Aeschlimann, Bernd Schimanski, and André Schneider. Please ensure that the full contributions of each author are acknowledged in the "Add/Edit/Remove Authors" section of our submission form.

- TM on page: 24.

4) We notice that your supplementary Figures are included in the manuscript file. Please remove them and upload them with the file type 'Supporting Information'. Please ensure that each Supporting Information file has a legend listed in the manuscript after the references list.

5) We note that your Data Availability Statement is currently as follows: "All relevant data are within the paper and its Supporting Information files". Please confirm at this time whether or not your submission contains all raw data required to replicate the results of your study. Authors must share the “minimal data set” for their submission. PLOS defines the minimal data set to consist of the data required to replicate all study findings reported in the article, as well as related metadata and methods (https://journals.plos.org/plosone/s/data-availability#loc-minimal-data-set-definition).

**Reviewers' Comments:**

Reviewer's Responses to Questions

**Part I - Summary**

Reviewer #1: In this study, Stettler et al. examined the assembly of TAC modules that reside within the mitochondrial outer membrane, using a combination of blue native PAGE and RNAi. Based on the results, the authors proposed a step wise assembly pathway from TAC40 oligomers, to larger TAC42-TAC40- TAC60 subcomplexes, and subsequent assembly with pATOM36-TAC65 subcomplex to form the OM TAC module. A detailed, accurate description of the OM-TAC module assembly is an important step to understand TAC biogenesis. Considering the limitation of BN-PAGE/immunoblots in subcomplex fractionation and detection (detailed below), additional approaches and controls are recommended to validate some of the key observations made in this study.

Reviewer #2: The manuscript by Stettler et al. investigates the composition of distinct subcomplexes formed by the tripartite attachment complex (TAC) components in Trypanosoma brucei. The authors identify detergent-soluble assemblies located in the outer membrane (OM) segment of the TAC. Their data suggest that OM TAC modules form three distinct subcomplexes: one containing TAC40, another with TAC42 and TAC60, and a separate subcomplex consisting of pATOM36 and TAC65. Through a combination of Blue Native PAGE (BNPAGE), RNA interference (RNAi)-mediated protein depletion, and restoration experiments in parasites lacking kinetoplast DNA (kDNA), the authors propose an assembly pathway that includes TAC40 oligomerization and subsequent binding events.

While the presented model aligns with the data, there are two main concerns regarding the robustness and functional significance of the findings.

**Part II – Major Issues: Key Experiments Required for Acceptance**

Reviewer #1: 1. Fractionation of OM complexes by BN-PAGE. In this study, BN-PAGE was combined with immunoblottings to detect different OM-TAC subcomplexes. Although a powerful method to study native protein complexes, BN-PAGE does have limitations. In Fig. 1A, in addition to the labelled 920, 770 and 535 subcomplexes, there are clearly other higher molecular weight bands (> 1,048kDa) in each of the blots for TAC40, TAC42-HA and TAC60-myc, suggesting the presence of higher order complexes containing all 3 subunits? Besides, it was concluded that the 920 KDa subcomplex contained much less TAC40 protein than the other, smaller subcomplexes (lines 175-176). This is an ambiguous statement. While it is possible that the 920 KDa complex is present at lower abundance, it is also possible that the transfer of the larger complexes is less efficient during the blotting process, leading to less intense signals compared to smaller complexes. As such, even larger complexes may be present but not detectable on BN-PAGE.

Considering the consistency issue often associated with BN-PAGE, and size-correlated protein transfer efficiency during blotting, it would be important to corroborate the BN-PAGE results with other complex fraction/detection methods that may be less affected by size. For example, size exclusion chromatography followed by mass spec or SDS-PAGE and immunoblots of the isolated subcomplexes, may help to verify the presence and relative abundance of different subcomplexes.

2. It appears that the tagged TAC42 and TAC60 subunits are introduced into different cell lines and thus the BN PAGE/immunoblot was performed separately for each subunit. This may explain the apparent size variation of the complexes on different blots. For example in Fig 1A, the two bands labelled as 920 and 770 complexes appeared to run much higher than what I assumed to be the 720 KDa reference band (not labeled) in the TAC60-myc blot, as compared to these bands in the TAC40 and TAC42-HA blots. Considering the variation in BN -PAGE results, and the importance of accurate subcomplex identification in this study, I would suggest expressing TAC42-HA and TAC60-myc tags in the same cell line, and probe for all 3 subunits (with different antibodies) on samples fractionated on the same BN-PAGE gel.

3. The presentation of Fig. 5 is confusing. Based on the gel/blot background, the NYsm sample (dotted lines) seems to be taken from a BN-PAGE experiment different to the other 3 samples. However, all 4 lanes are in the same solid line box and share the same molecular weight standard. It seems that the myc tag on TAC40 significantly increased the size of each of the subcomplexes? Please clarify this point. Besides, was the blot probed with anti-TAC40? Based on Fig. S3, ani-TAC40 has a nonspecific band in denatured samples. Nonspecific label seems absent under native conditions, at least for the BSF samples shown in Fig. 5. But how about PCF cells? Could any of the TAC40-containing subcomplexes observed in Figs 1-4 be caused by non-specific labelling? Think this is an important control to include.

Reviewer #2: Reliance on a Single Methodology: The majority of the conclusions rely on BNPAGE analyses. Incorporating additional techniques, such as mass spectrometry of affinity-purified complexes or reconstitution experiments in a heterologous system, would significantly strengthen these findings.

Functional Clarity: The precise role of the identified TAC subcomplexes remains unclear. Clarifying their functional relevance would greatly enhance the significance and impact of this study.

**Part III – Minor Issues: Editorial and Data Presentation Modifications**

Reviewer #1: 1. As BN PAGE results may vary greatly between experiments and gel conditions, the BN PAGE methods should be described in greater details. Is there a reference for the BN-PAGE protocol used in this study? What was the buffer used to prepare for the 4-13% gel? What was the running buffer? And the running conditions?

2. Lines 242-243. Depletion of either pATOM36 or TAC65 disrupted the 440 complex. This is not unexpected for protein complexes, where depletion of a subunit leads to the dissociation of the entire complex. This however, does not mean that the 440 complex contains only these two subunits (as concluded by the authors). The presence of other subunits in this complex cannot be excluded based on this experiment alone.

3. Line 95. “a few thousand copies have [been] identified so far…”

4. The 535KDa subcomplex was found to be TAC40 oligomer (lines 237-238). Could the authors explain a bit more about this? The 535 subcomplex could be labelled by anti-TAC40, but not by anti-myc or anti-HA (for TAC60 and TAC42 respectively), but other protein subunits could still be present in this subcomplex?

Reviewer #2: Figure 1B: It is unclear whether the molecular mass markers differ between the two blots presented. In the absence of mass spectrometry for affinity- or immuno-purified complexes separated by size, the conclusion that the 535 kDa complex consists solely of TAC40, and the 770 kDa and 920 kDa complexes contain varying amounts of TAC42 and TAC60, is speculative. This conclusion needs further support beyond BNPAGE results. Additionally, the authors should emphasize that co-immunoprecipitation (co-IP) indicates protein interaction but does not confirm direct protein-protein interactions. Furthermore, RNAi efficiency should be explicitly assessed and presented.

Figure 2: Clarify the meaning of the '+' and '–' symbols in the figure; presumably, these refer to RNAi induction status. Additionally, specify the duration of RNAi induction and quantify the efficiency of targeted protein depletion. The appearance of a TAC42-containing smear in the TAC40 and TAC60 RNAi conditions appears to be overstated and needs cautious interpretation.

Figure 7A: Mass spectrometry analysis of the pull-down complexes would substantially reinforce the proposed model of complex architecture and provide direct evidence supporting the subcomplex formation.

PLOS authors have the option to publish the peer review history of their article (what does this mean? ). If published, this will include your full peer review and any attached files.

**Do you want your identity to be public for this peer review?** For information about this choice, including consent withdrawal, please see our Privacy Policy .

Reviewer #1: No

Reviewer #2: No

**Figure resubmission:**

**Reproducibility:**



---

## [Decision Letter · Decision Letter 1]

29 Aug 2025

Dear Prof Schneider,

We are pleased to inform you that your manuscript 'Assembly of the mitochondrial outer membrane module of the trypanosomal tripartite attachment complex' has been provisionally accepted for publication in PLOS Pathogens.

Best regards & HAPPY RETIREMENT !!

Dominique 

Dominique Soldati-Favre

Section Editor

PLOS Pathogens

Sumita Bhaduri-McIntosh

Editor-in-Chief

PLOS Pathogens

orcid.org/0000-0003-2946-9497

Michael Malim

Editor-in-Chief

PLOS Pathogens

orcid.org/0000-0002-7699-2064

Reviewer Comments (if any, and for reference):

Reviewer's Responses to Questions

**Part I - Summary**

Reviewer #1: Most of my questions were addressed. I have no further comments.

Reviewer #2: The main strength of the study is the element of discovery that rationalizes previous findings and, most importantly, provides a path to deeper understanding of TAC assembly and function at the structural level. Additional analysis, along with many meaningful edits presented in the revision, outweighs my main concern about relying on a single approach to draw substantive conclusions. I have no further comments on the manuscript.

**Part II – Major Issues: Key Experiments Required for Acceptance**

Reviewer #1: (No Response)

Reviewer #2: None.

**Part III – Minor Issues: Editorial and Data Presentation Modifications**

Reviewer #1: (No Response)

Reviewer #2: None.

PLOS authors have the option to publish the peer review history of their article (what does this mean? ). If published, this will include your full peer review and any attached files.

**Do you want your identity to be public for this peer review?** For information about this choice, including consent withdrawal, please see our Privacy Policy .

Reviewer #1: No

Reviewer #2: No

---

## [Editor Report · Acceptance letter]

Dear Prof Schneider,

We are delighted to inform you that your manuscript, "Assembly of the mitochondrial outer membrane module of the trypanosomal tripartite attachment complex," has been formally accepted for publication in PLOS Pathogens.

Best regards,

Sumita Bhaduri-McIntosh

Editor-in-Chief

PLOS Pathogens

orcid.org/0000-0003-2946-9497

Michael Malim

Editor-in-Chief

PLOS Pathogens

orcid.org/0000-0002-7699-2064